# Not Just a Flash in Time: Interpreting Long Event Streams through Language

## Abstract

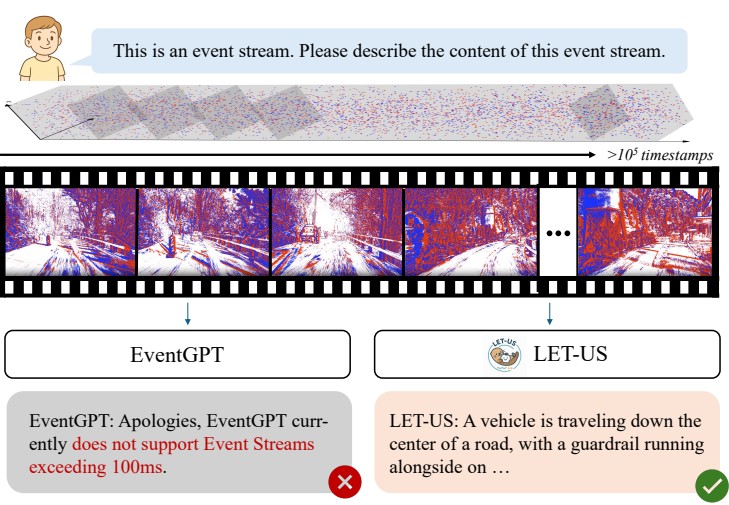

Figure 1: Comparison with state-of-the-art models.

Event cameras operate asynchronously with microsecond-level temporal precision and generate sparse event streams, enabling low-latency visual perception under high dynamic range conditions. However, current multimodal large language models (MLLMs) remain suboptimal when handling such data: they either fail to effectively interpret event streams or are limited to very short temporal sequences. To address this problem, we propose a unified approach for long event-stream–text understanding. This method employs an adaptive compression mechanism that significantly reduces input volume while preserving key motion and structural cues, thereby supporting long-term cross-modal reasoning. The training pipeline adopts a two-stage optimization process: the model is first guided to develop representational capacity for streaming data, followed by cross-modal alignment to enhance semantic consistency between event and textual modalities. To handle the substantial temporal information inherent in long event streams, the model uses text-guided cross-modal queries to select salient features and combines hierarchical clustering with similarity scoring to extract representative event segments. During training, a large-scale event–text aligned dataset is curated and constructed, facilitating more effective embedding of event features within the semantic space of language models. In addition, we establish a comprehensive benchmark covering a diverse set of tasks including reasoning, captioning, classification, temporal localization, and moment retrieval. Experimental results demonstrate that the proposed approach outperforms existing state-of-the-art MLLMs in both descriptive accuracy and semantic understanding on long-duration event streams. All datasets, code, and models will be released publicly.

# 1 INTRODUCTION

Event cameras' extreme sensitivity to illumination changes lets them capture dynamic scenes and complex lighting conditions (Chakravarthi et al., 2024; Gallego et al., 2020; Gehrig & Scaramuzza, 2024; Kudithipudi et al., 2025; Wang et al., 2024; Zhou et al., 2023). They've proven effective in high-speed motion and challenging lighting, drawing growing interest (Zheng et al., 2024). Key tasks include object detection (Yang et al., 2025), tracking (Apps et al., 2025), 3D reconstruction (Feng et al., 2025), and high-level scene understanding (Kong et al., 2024).

Multimodal Large Language Models (MLLMs) (Zhang et al., 2024a) excel at handling both visual and textual data. To prevent inference degradation with long video inputs, several methods have been introduced (Song et al., 2024; Wang et al., 2025; Zhang et al., 2024b). Many highlight that videos contain abundant redundant, information-sparse tokens (Choudhury et al., 2024), which can impede long-video understanding (Cheng et al., 2024).

However, existing research focuses on RGB video inputs. Multimodal models for event streams are still emerging and have not addressed the inference performance and efficiency challenges of long-duration event data. Event streams feature much longer temporal sequences and higher noise levels: one second can produce about $10^6$ timestamps, so even a few seconds yield extremely long sequences, which undermining MLLMs' robustness. From a data perspective, text-annotated event videos are extremely rare, impeding progress in event understanding. Moreover, current event datasets are limited to short-span driving scenarios and are insufficient for comprehensive event-video studies.

To investigate how models learn from feature tokens, we analyze both the training stage (with particular focus on token-level loss computation) and the inference stage (examining how feature tokens participate in inference computations). We evaluate the performance of various token-reduction strategies and explore their robustness when applied to event data. Our findings reveal that the distribution of visual information in event streams across the temporal dimension is nonuniform. For example, in event streams spanning millions or hundreds of millions of time-stamped events, bursts of information may occur only in a few temporal segments, while in other segments no information is produced. Fixed-interval sampling or random sampling–based trimming strategies often allocate excessive attention to information-empty segments, neglecting the crucial segments where information is highly concentrated.

Motivated by these observations, we propose LET-US for long event-text understanding of scenes, which is a framework designed to preserve as much useful input information as possible while eliminating redundancy in long event streams, thus adapting to the context-length limits of prevalent large language models. LET-US leverages semantic cues from a given prompt to guide the model in selecting the event segments of interest and employs a clustering-based approach to dynamically identify the most representative tokens. To bridge the substantial gap between the event-stream and text modalities, we adopt a two-stage fine-tuning paradigm. In the first stage, we pre-train on conventional RGB datasets to enhance the model's capacity for streaming data. In the second stage, we fine-tune our model on a public, large-scale event-frame image dataset to boost its semantic understanding of event streams, since there are no large-scale public event stream–text alignment datasets and creating them is costly. We posit that the contour and representation information contained in event frames facilitate transferring the model's RGB modality understanding to the event-stream modality. Consequently, even training solely on datasets composed of event frames effectively enables the model to learn diverse object semantics within the event-stream modality, thereby laying a robust semantic foundation for subsequent event-stream scene understanding.

Finally, to address the severe scarcity and limited diversity of text-annotated event data, we leveraged existing models to generate a large-scale, diverse Event-Image-QA-1M (EIQA-1M) dataset comprising over one million question-answer pairs for fine-tuning. To further evaluate our model's capability in long-event-stream understanding, we constructed Event-Video-QA-Benchmark (EVQA-Bench), a benchmark of over 50K question-answer pairs covering classification, reasoning, moment retrieval and temporal localization, and captioning tasks, with timestamp spans ranging from $10^5$ to $10^9$. EVQA-Bench fills the current void in datasets for long-event-stream semantic understanding. Empirical experiments demonstrate that LET-US outperforms other mainstream models across these tasks. We show our demo for understanding a long event stream in Figure **??**.

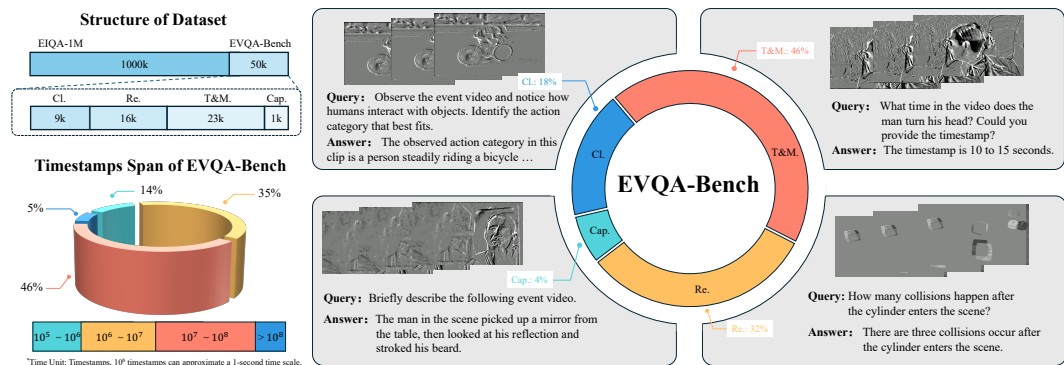

Figure 2: Dataset Overview, including EIQA-1M and EVQA-Bench. EVQA-Bench encompasses classification (CL.), captioning (Cap.), reasoning (Re.), temporal localization and moment retrieval (T&M) tasks.

Through comprehensive experiments, we demonstrate LET-US outperforms existing approaches in long-sequence reasoning, question answering, and temporal localization on event streams. Our contributions are as follows:

- We propose LET-US, the first framework to explore textual alignment and language understanding for long event streams, which can achieve the comprehension of event streams lasting up to $10^9$ timestamps.

- We design a sequence reduction approach tailored for long event streams, which selects salient segments through cross-modal guidance and further condenses event stream information via clustering methods. This approach compresses the long event-stream sequence to reduce the inference cost of the LLM, while preserving as much of the holistic information from the segments of interest as possible.

- We construct a large-scale event–text dataset EIQA-1M for training, comprising over 1M QA samples to facilitate direct alignment of event and text representations. We also develop a benchmark for evaluating long event streams, namely EVQA-Bench, where the longest stream lasts over $10^9$ timestamps. Based on EVQA-Bench, LET-US outperforms prior models by a substantial margin.

## 2 RELATED WORK

**Event Understanding.** The event modality has recently attracted considerable attention due to its extremely high temporal resolution and its ability to capture high dynamic range scenes. Some research institutions collected a series of event data using an event camera Gehrig et al. (2021); Rebecq et al. (2019). In addition, Eventbind Zhou et al. (2024) aligned event, image, and text modalities, thereby bridging the gap between events and other modalities. EventGPT Liu et al. (2025) extended multimodal large language models to include the event modality, achieving the understanding of event stream videos with $10^5$ timestamps (100ms, according to their code[1].). However, all of these methods remain limited by the relatively small amount of event stream information they process.

**RGB-based long-sequence understanding.** Some works focus on channel pruning within Transformer architectures, analyzing ViT components and assigning importance scores to determine which feature dimensions to prune. SAViT (Structure-Sware Vision Transformer) Zheng et al. (2022) explores dependencies among feature dimensions to guide pruning more effectively. GOHSP (Graph and Optimization-based Heterogeneous Structured Pruning) Yin et al. (2023) measures head importance using a Markov chain–based inter-head relationship graph and applies soft-pruning masks to adjust feature channels during training. While these methods are robust, they require extensive fine-tuning. Other studies prune the input token sequence: token pruning sparsifies the patch sequence by removing less important patches or merging tokens, retaining those most attended by the CLS

---
[1]https://github.com/XduSyL/EventGPT

token. ToMe Bolya et al. (2022) merges tokens and shows strong results at inference time. Inspired by ToMe's success, some methods use hierarchical clustering for further merging. However, these efforts focus on short token sequences and a single modality. A series of works also target long-video understanding: RLT (Run-Length Tokenization) Choudhury et al. (2024) uses run-length positional encoding to mark and remove "static" patches, while LongVU (Long Video-Language Understanding) Shen et al. (2024) applies text-guided frame-level filtering to down-sample frames. Though impressive, these methods are designed for RGB video and struggle with sparse, noisy event data. Most rely on fixed windows for token reduction. Therefore, we introduce new training techniques and token-pruning methods to better leverage large-scale event datasets.

## 3 DATASET GENERATION

Event-based datasets have been widely studied in computer vision tasks. However, large-scale open-source event-text pair datasets for training multimodal large language models remain scarce. To address this gap, we developed a pipeline that automatically generates and filters high-quality event-text data by combining an existing event simulator Hu et al. (2021) with the ChatGPT model, alongside human-in-the-loop refinement. Using this pipeline, we constructed a dataset of over one million event-photo and event-stream pairs aligned with text: EIQA-1M and EVQA-Bench. The distribution of our datasets is shown in Figure 2.

### 3.1 EIQA-1M

We first annotated text-based QA pairs on an existing event-image dataset. ImageNet Deng et al. (2009) is the foundational benchmark for visual recognition tasks. N-ImageNet Kim et al. (2021) was aligned to ImageNet images using an event simulator. We then paired N-ImageNet images with QA annotations to form N-ImageChat. Please refer to *appendix* for more details.

### 3.2 EVQA-BENCH

We develop a suite of leading video datasets and, using our custom-built processing pipeline, generated a high-quality event–text corpus of over 50K examples. This collection spans content domains such as autonomous driving, cinematic narratives, and human actions, and supports a variety of tasks, including classification, captioning, QA, conversational dialogue, temporal localization, and reasoning. Specifically, the classification task is subdivided into two parts: (1) human action recognition, covering over 1000 items; (2) object classification built upon the existing N-Caltech101 Orchard et al. (2015) dataset, which includes more than 8,000 samples. For the captioning task, to ensure that evaluation spans event streams of different durations, we define two subcategories: sparse event streams (approximately $3 \times 10^6$ in timestamp span) and dense event streams (timestamp spans exceeding $10^8$).

## 4 LET-US

### 4.1 OVERVIEW

LET-US is an event-based MLLM capable of understanding and generating responses grounded in event data. We provide an overview in Figure 3. LET-US formulates event-data processing as an event-driven approach tailored for question-answering and descriptive generation tasks. By leveraging the high temporal resolution and extended dynamic range of event streams, LET-US significantly enhances the comprehension of scenes traditionally challenging for standard visual models, such as those under low-light conditions or involving high-speed motion, and extends this capability to event streams spanning longer timestamps.

We partition continuous event streams into discrete temporal windows based on timestamp intervals, aggregating each window into a bin. Mathematically, an event bin within a time window can be represented as a quadruple $S = x_i, y_i, t_i, p_i$, where each event comprises spatial coordinates $(x_i, y_i)$, timestamp $t_i$, and polarity $p_i$. Inspired by Cambrian's Tong et al. (2024) exploration of integrating SigLIP and DINOv2 to enhance performance on vision tasks, we leverage the feature extraction capabilities of SigLIP2 Zhai et al. (2023) and DINOv2 Oquab et al. (2023) to initialize our event encoder. We firstly use an event encoder to extract features from the event data, capturing

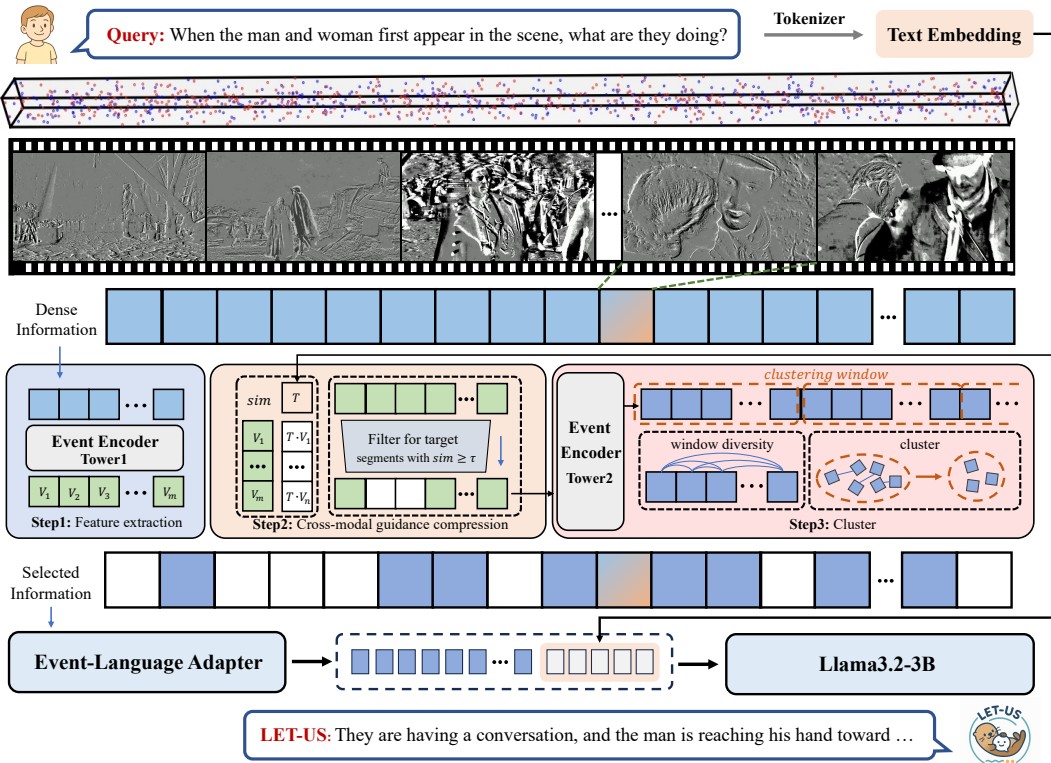

Figure 3: Overview of LET-US. After segmenting the event stream into bins, features are extracted from each bin and their similarities with the query text are calculated to identify relevant bins. The remaining bins undergo hierarchical clustering for further reduction. Finally, features from these reduced event bins are processed through an Event-Language Adapter, concatenated with the query features, and fed into Llama for answer generation.

local spatiotemporal characteristics, and then generate a more comprehensive representation using a spatiotemporal encoder. Next, a two-stage projector aligns event and text features, and the response is generated through a large language model. The model leverages the query prompt and the distribution of sequence information to perform compression, providing an efficient method for understanding event streams. LET-US is a framework suitable for event streams with varying temporal spans.

## 4.2 INFORMATION COMPRESSION

**Cross-Modal Guided Compression.** Drawing inspiration from the work of LongVU Shen et al. (2024), we adopt siglip2-so400m-patch14-384 as part of our event encoder to encode each temporal bin $E_t$, obtaining event-encoded features $V = \{V_1^S, \ldots, V_T^S\} \in \mathbb{R}^{T \times (H_h W_h) \times D_v}$ where $H_h \times W_h$ represents the spatial dimensions of the event bin features, and $D_v$ denotes the channel dimension. Clearly, the obtained features can hardly be considered as closely related to our content of interest. Therefore, we introduce interactions between textual and event modalities, where the user's textual query is input into the model to guide the initial compression of the feature sequence. Specifically, we calculate the cosine similarity between each bin and the query text vector, selecting bins whose similarity exceeds the threshold $\tau$. This process can be formulated as:

$$\widetilde{V} = \{V_t^S \mid \text{sim}(V_t^S, q) \geq \tau, \ t = 1, 2, \ldots, T\}, \tag{1}$$

where $q$ denotes the provided query text, $V^t$ represents the features of the event bin, and $\tau$ denotes the predefined similarity selection threshold. After cross-modal guided filtering, we obtain a sequence of $k$ bins containing the content of our interest, represented as $B = \{\text{bin}_1, \ldots, \text{bin}_k\}$.

**Temporal Compression.** The information represented by event streams is highly non-uniformity. Simply feeding the entire sequence or selecting segments at fixed intervals is evidently not a wise choice, particularly when the sequence contains tens of millions of timestamps. We leverage the powerful feature extraction capability of DINOv2 to compute features for the extracted bins, obtaining $V' = \{V_1^D, \ldots, V_k^D\}$. Within each non-overlapping window containing $J$ bins, we treat each bin as an individual data point and perform a density-driven clustering and aggregation. This process can be described as follows: Partition $V'$ into $M = \lfloor \frac{k}{J} \rfloor$ non-overlapping windows of size $J$ each:

$$W_m = \{V_{(m-1)J+1}^D, \ldots, V_{mJ}^D\}, \quad m = 1, \ldots, M. \tag{2}$$

Within each window $W_m$, we firstly $\ell_2$–normalize all vectors, then compute the average pairwise cosine distance:

$$D_m = \frac{2}{J(J-1)} \sum_{1 \le i < j \le J} \left[1 - \langle V_i, V_j \rangle\right]. \tag{3}$$

Here $\langle V_i, V_j \rangle$ is the cosine similarity; $D_m \in [0, 2]$ quantifies how dense the information in $W_m$ is.

Rather than fixing distance-thresholds, we map the diversity $D_m$ directly to an integer cluster count $R_m \in [1, J]$:

$$R_m = \max\left(1, \min\left(J, \text{round}\left(\frac{D_m}{2} J\right)\right)\right). \tag{4}$$

Thus when $D_m = 0$ (all bins nearly identical), $R_m = 1$; when $D_m = 2$ (maximally diverse), $R_m = J$; intermediate $D_m$ yields proportional cluster counts.

On the $J$ normalized vectors in $W_m$, perform bottom-up average-linkage clustering to partition them into exactly $R_m$ disjoint clusters:

$$\{C_{m,r}\}_{r=1}^{R_m} = \text{HAC}_{\text{avg}}\left(\{V_{(m-1)J+1}^D, \ldots, V_{mJ}^D\}, R_m\right), \tag{5}$$

where $\text{HAC}_{\text{avg}}(X, K)$ denotes performing bottom-up average-linkage hierarchical clustering on the set X and partitioning it into exactly $K$ disjoint clusters.

For each cluster $C_{m,r}$, compute its new-bin feature as the arithmetic mean of its member-bin vectors:

$$\hat{V}_{m,r}^D = \frac{1}{|C_{m,r}|} \sum_{i \in C_{m,r}} V_{(m-1)J+i}^D, \quad r = 1, \ldots, R_m. \tag{6}$$

Finally, concatenate all aggregates in temporal order to obtain the compressed token sequence as follow:

$$\{\hat{V}_{1,1}^D, \ldots, \hat{V}_{1,R_1}^D, \ldots, \hat{V}_{M,1}^D, \ldots, \hat{V}_{M,R_M}^D\}. \tag{7}$$

### 4.3 TRAINING PIPELINE

Considering the substantial gap between event and textual modalities, we designed a pre-training strategy to initialize our framework. Our pre-training consists of two phases: visual-language training and event-language training. This phased approach helps achieve efficient cross-domain modal alignment, enhancing LET-US's capability for event comprehension and reasoning.

**Visual-language training.** Compared to the event–text task, vision–text learning has larger datasets and more mature alignment methods, so we first align vision and text via two steps: image–language and video–language. This builds foundational scene understanding and enhances the model's ability to grasp streaming data.

**Event-language training.** We fine-tune on our custom EIQA-1M dataset to align event streams with natural language, thereby enhancing the model's spatiotemporal reasoning and descriptive capabilities. We contend that EIQA-1M composed exclusively of event frames can effectively improve the model's semantic understanding of distinct objects in the event-stream modality, laying a semantic foundation for subsequent event-stream comprehension.

## 5 EXPERIMENT

**Implementation Details.** To balance scene understanding capability with lightweight deployment, we adopt Llama3.2-3B as our backbone, demonstrating proposed method's ability to comprehend long event streams on a small model. The model is trained on eight NVIDIA A100-SXM4-80 GB GPUs. In the video–language training phase, we employ the LLaVA-OneVision Li et al. (2024) and VideoChat2-IT Li et al. (2023) datasets in a two-stage procedure for one epoch with a batch size of 64; the learning rate is set to $10^{-5}$ with a warm-up ratio of 0.03. For the event–language fine-tuning phase, we use our self-constructed EIQA-1M dataset to train for one epoch (batch size = 64), again with a learning rate of $10^{-5}$ and warm-up ratio of 0.03. During this stage, each event stream is compressed via our proposed adaptive compression method ($\tau = 0.5$, $J = 8$).

**Comparison Methods.** To the best of our knowledge, no existing model can process event streams with timestamp spans beyond $10^5$. EventGPT handles up to $1 \times 10^5$ timestamps, corresponding to event videos no longer than 100 ms. Therefore, to enable comparison, we selected the N-Caltech101 dataset with the fewest timestamps and were forced to truncate it to within $1 \times 10^5$ timestamps to satisfy EventGPT's input limitation. To enable fair comparison with mainstream video-understanding models, we generate corresponding RGB videos for all event-based benchmark datasets so that they meet the input requirements of these models. We use the model's accuracy in each task as the evaluation metric.

### 5.1 COMPARISON WITH STATE-OF-THE-ART MODELS

**Quantitative Results.** Table 1 presents our experimental accuracy results across various event understanding tasks. Our method outperforms all baseline models including Video-ChatGPT Maaz et al. (2023), Chat-UniVi Jin et al. (2024), Video-LLaVA Lin et al. (2023), LongVU Shen et al. (2024), VideoLLaMA3 Zhang et al. (2025), Qwen2.5-VL Bai et al. (2025) and EventGPT Liu et al. (2025). on these benchmarks, despite having fewer parameters. Specifically, in classification tasks, our model surpasses existing models by at least 3% accuracy in human action recognition. In object classification (N-Caltech101), it achieves comparable performance to the 8B-parameter VideoL-LaMA3, while exceeding other models by at least 12% accuracy. Even on the classification task of the low-timestamp N-Caltech101 dataset, it still outperforms EventGPT. Simultaneously, it achieves state-of-the-art performance in other tasks. The experiments demonstrate that our model consistently achieves state-of-the-art performance in classification, reasoning, moment retrieval, and captioning tasks across event streams of various timestamp spans.

| Models | LLM Backbone | Params | Encoder | Classification | | Rea-soning | T&M | Captioning | |
|---|---|---|---|---|---|---|---|---|---|
| | | | | Action | N-Caltech101 | | | Sparse | Dense |
| Span (order of magnitude) | | | | $10^6$ | $10^5$ | $10^6$ | $10^6 \sim 10^9$ | $10^6$ | $10^8$ |
| Video-ChatGPT | Vicuna–v1.5 | 7B | CLIP ViT–L/14 | 0.25 | 0.25 | 0.25 | 0.18 | 0.26 | 0.20 |
| Chat-UniVi-7B | Vicuna–7B | 7B | CLIP ViT–L/14 | 0.37 | 0.58 | 0.31 | 0.27 | 0.32 | 0.20 |
| Video-LLaVA | Vicuna–7B | 8B | CLIP ViT–L/14 | 0.31 | 0.31 | 0.30 | 0.18 | 0.29 | 0.37 |
| LongVU | Qwen2–7B | 7B | DINOv2+SigLIP | 0.30 | 0.36 | 0.39 | 0.26 | 0.27 | 0.25 |
| VideoLLaMA3 | Qwen2.5–7B | 8B | SigLIP | 0.41 | 0.70 | 0.41 | OOM | 0.22 | OOM |
| Qwen2.5-VL | Qwen2.5-VL-7B | 7B | Reengineered ViT | 0.30 | 0.40 | 0.30 | OOM | 0.43 | OOM |
| EventGPT | Vicuna–v1.5 | 7B | OpenCLIP ViT-L/14 | ✗ | 0.40 | ✗ | ✗ | ✗ | ✗ |
| **LET-US (ours)** | Llama3.2-3B | 3B | SigLIP2+DINOv2 | **0.44** | **0.70** | **0.42** | **0.35** | **0.49** | **0.40** |

Table 1: Comparison with State-of-the-Art Models. T&M represents Temporal Localization and Moment Retrieval. The span is measured in timestamps. OOM indicates the model cannot complete the task within 80 GB CUDA memory. ✗ indicates the model does not support that input length.

**Qualitative Results.** To demonstrate the unique advantages of event streams over conventional RGB video and to highlight our method's capability in understanding long event streams, we provide qualitative results in Figure 4 and Figure 5. Figure 4 illustrates examples of LET-US's performance across different tasks. In Figure 5, we present examples of performance on real-world event datasets. The event streams here using red and blue colors to represent positive and negative polarities, respectively. Specifically, Figure 5a employs the DSEC Gehrig et al. (2021) dataset to evaluate the scene understanding capabilities of LET-US. Figure 5b illustrates that, in scenes with high dynamic

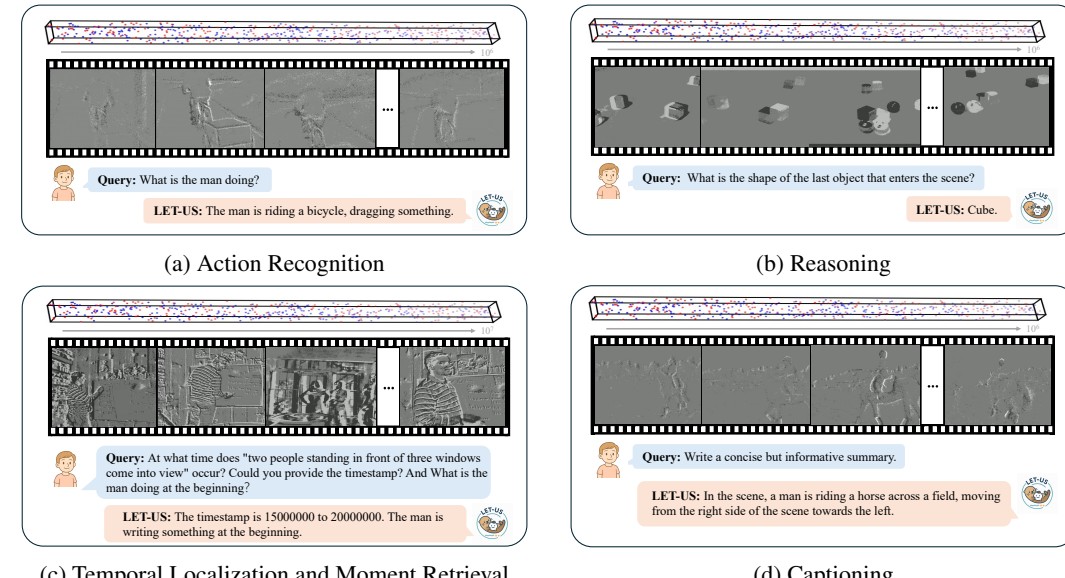

(a) Action Recognition  (b) Reasoning

(c) Temporal Localization and Moment Retrieval  (d) Captioning

Figure 4: Examples of LET-US's performance across different tasks.

range lighting, conventional RGB vision fails to accurately perceive surrounding objects due to significant loss of visual information. In contrast, event streams effectively capture information from regions that are overexposed or underexposed in traditional RGB frames, thus enabling accurate environmental perception under challenging lighting conditions. Figure 5c compares the perception capabilities of RGB cameras and event cameras for moving objects. The results clearly indicate that event streams significantly outperform traditional cameras in dynamic scenes with complex lighting.

## 5.2 ABLATION STUDIES

To dissect the impact of each architectural component on performance, we first establish a baseline model that mirrors the LLaMA architecture, undergoes vision–language pre-training and fine-tuning, and selects information from long event streams via random sampling. *Variant A* substitutes this random strategy with fixed-interval sampling that shares all the same parameters with LET-US; the baseline and *Variant A* further integrate query-guided cross-modal compression, while *Variant B* omits this module. All methods saturate the model's context window with as much input as its length permits. As reported in Table 2, our proposed LET-US framework surpasses the baseline and both variants across all evaluation tasks. In Dense Captioning, where the input sequences are extremely long, hierarchical clustering demonstrably outperforms sampling schemes that ignore the distribution of salient events. For Temporal Localization and Moment Retrieval, the anticipated benefit of query-guided compression is confirmed, yielding markedly higher localization accuracy. Notably, random and fixed-interval sampling combined with query guidance yield performance close to or even matching LET-US, implying that, within temporal localization settings, the query-driven cross-modal guidance contributes more decisively to performance than the specific sampling policy itself.

| Mode Type | Sampling | CMGC | Task | |
| --- | --- | --- | --- | --- |
| | | | T&M | DC |
| Baseline | Random | ✓ | 0.35 | 0.28 |
| *Variant A* | Interval | ✓ | 0.33 | 0.25 |
| *Variant B* | Cluster | ✗ | 0.29 | 0.31 |
| **LET-US (ours)** | **Cluster** | ✓ | **0.35** | **0.40** |

Table 2: Component Analysis. T&M represents Temporal Localization and Moment Retrieval. CMGC represents Cross-Modal Guidance Copression. DC represents Dense Captioning.

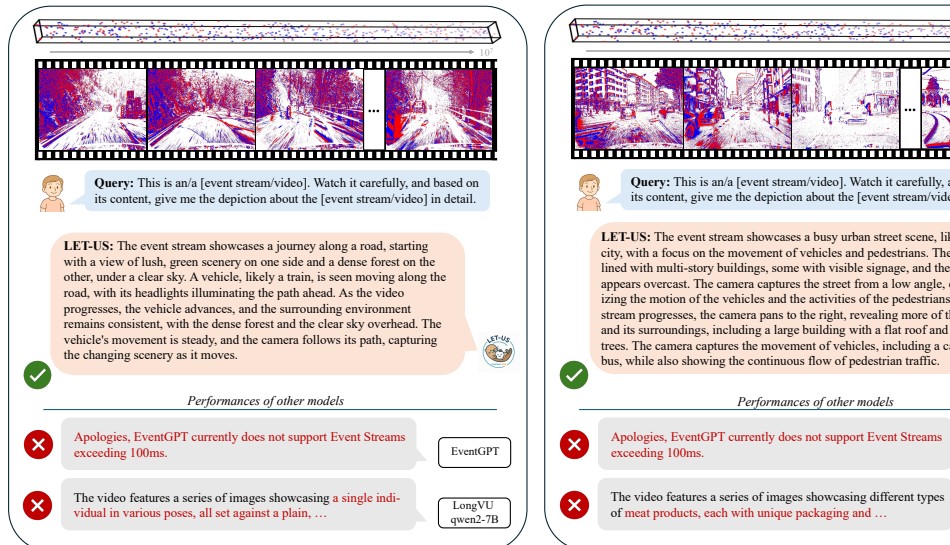

(a) Scene understanding performance on real datasets captured by event cameras. [event stream/video] indicates that we input the token "event stream" for LET-US and EventGPT, and the token "video" for video understanding models.

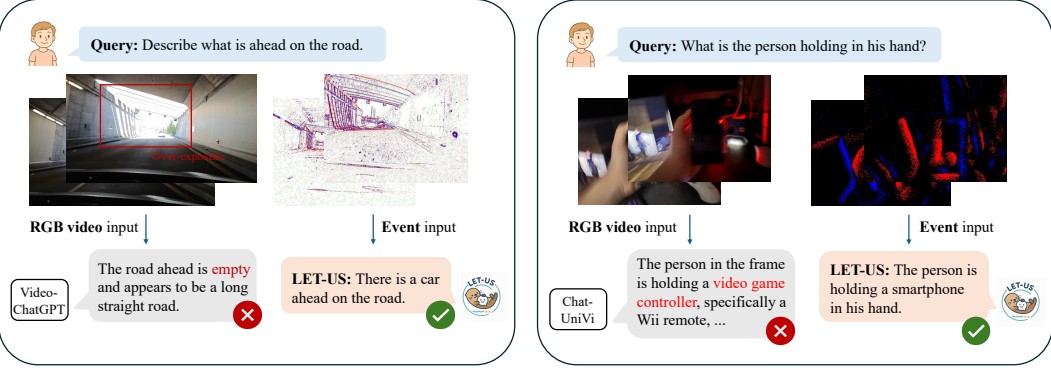

(b) Information loss caused by poor lighting.   (c) Information loss caused by blurring.

Figure 5: Understanding performance on real-world datasets captured by event cameras.

# 6 CONCLUSION

In this paper, we introduce LET-US, the first Multimodal Large Language Model (MLLM) to explore semantic alignment and understanding of long event streams. To this end, we proposed a cross-modal guidance compression method to select informative segments from event streams, followed by a clustering-based method to further mitigate token redundancy. This approach enables a significant reduction in the input volume of event data while maximizing the preservation of relevant semantic information. To address the modality gap between event data and text, we construct two event-text alignment datasets: EIQA-1M and EVQA-bench. The EIQA-1M dataset supports model training and fine-tuning via a two-stage paradigm, while EVQA-benc serves as a dedicated benchmark for comprehensive evaluation of model capabilities. Experimental results on multiple event understanding benchmark tasks consistently validate the superior performance of our proposed model.

**Limitation and future work.** Please refer to appendix for limitations and future works.

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
