# OpenReview forum: "Not Just a Flash in Time: Interpreting Long Event Streams through Language"
_ICLR.cc/2026/Conference — ICLR 2026 Conference Withdrawn Submission_

### Official Review · Reviewer_Jz4q · 2025-10-23

**Soundness:** 3
**Presentation:** 2
**Contribution:** 3
**Rating:** 6
**Confidence:** 4

**Summary:**

The authors introduce LET-US, a framework designed for understanding long event streams, which achieves the comprehension of event streams with up to 10⁹ timestamps, and construct EVQA-1M and EVQA-Bench for training and evaluation.

**Strengths:**

1. Proposed EVQA-1M and EVQA-Bench, contributing to the training and evaluation of event-based vision MLLMs.
2. Introduced LET-US, a framework for understanding long event streams, extending event stream comprehension from 10⁵ to 10⁹ timestamps.
3. Designed a sequence reduction approach tailored for long event streams, which selects salient segments through cross-modal guidance and further condenses event information via clustering methods, enabling effective compression of long event streams.

**Weaknesses:**

1. Insufficient experimental validation. The authors claim to propose a framework for long event streams with compression and acceleration, yet they fail to analyze key performance metrics such as TTFT (Time-To-First-Token) and throughput.
2. Incomplete related work analysis. The discussion overlooks existing event-based vision MLLMs such as EventVL and LLaFEA. Although these models fuse frame and event streams, they are still relevant and should be discussed for a more comprehensive comparison.
3. Lack of analysis on the EVQA-1M dataset. While the authors introduce the EVQA-1M dataset, they do not analyze its data sources, such as which datasets it is derived from.
4. Fairness of experiments. The authors should clarify whether N-Caltech101 or similar datasets were included in the training data, to ensure that the comparison with EventGPT in the main experiments is conducted under zero-shot conditions for fair evaluation.

**Questions:**

See weaknesses.

---

### Official Review · Reviewer_ERew · 2025-10-30

**Soundness:** 2
**Presentation:** 2
**Contribution:** 3
**Rating:** 2
**Confidence:** 3

**Summary:**

This paper aims to address the problem that existing MLLMs cannot effectively parse event streams and can only handle short temporal sequences. The main contributions include:
1. Proposing LET-US, a method for long event stream-text understanding that can comprehend event streams with up to 10^9 timestamps.
2. Designing a compression method for long event sequences: filtering important segments by referencing cross-modal information, then further compressing event information through clustering.
3. Constructing EIQA-1M, an event-text alignment training dataset, and EVQA-Bench, a long event stream evaluation benchmark.

**Strengths:**

This paper is the first to explore methods for enhancing MLLMs in understanding long event streams. The authors leverage the characteristic of uneven information distribution in event streams and propose an information density-driven dynamic clustering strategy, combined with text query-based event segment filtering. The innovation of this work is primarily reflected at the application level—integrating existing techniques to solve practical problems rather than proposing entirely new algorithms. From a technical perspective, the method design is systematic and complete, with significant performance improvements. However, the ablation studies are insufficient, lacking independent analysis of key components. Additionally, the paper does not release code or data, and hyperparameter settings are not sufficiently detailed, which affects the reproducibility of the work. The paper structure is clear with excellent visualizations, but implementation details need to be supplemented. The dataset itself is an important contribution with broad application prospects.

**Weaknesses:**

1.The paper aims to address the insufficient understanding of long-sequence event streams by existing MLLMs. However, the temporal compression scheme adopted in this work has inherent limitations: when event streams contain dense and highly heterogeneous information, the compression mechanism inevitably loses fine-grained information, making it difficult to achieve truly lossless processing of long sequences.
2.The paper primarily compares against a limited number of event processing models such as EventGPT and RGB conversion baselines, but lacks comparisons with other non-LLM baselines specifically designed for processing long-sequence event streams in recent years, such as pure event stream models based on recurrent neural networks or attention mechanisms. The absence of these comparative experiments affects the comprehensive evaluation of the method's advantages.
3.The paper lacks quantitative analysis on real-world scenario datasets, including key metrics such as model performance, inference latency, and resource consumption, which are crucial for evaluating the practical value of the method.

**Questions:**

1.The ablation study only covers the T&M and DC components, with insufficient contribution analysis of other key modules (such as the "two-stage training pipeline" and "Event-Language Adapter"). How were the critical hyperparameters (such as similarity thresholds, window sizes, etc.) selected? It is recommended to demonstrate through sensitivity analysis experiments that these are the optimal parameters.
2.The paper mentions that dataset construction relies on large language models combined with manual review, but does not adequately explain the specific implementation process, quality control mechanisms, degree of human annotation involvement, and other details.
3.There is a citation error "Figure ??" at line 107 that needs to be corrected.
4.Using only accuracy as the evaluation metric is too coarse-grained. It is recommended to supplement with more detailed performance analysis metrics such as precision, recall, F1 score, and confusion matrix to comprehensively evaluate the model's performance across different categories.

---

### Official Review · Reviewer_C5Fv · 2025-10-31

**Soundness:** 2
**Presentation:** 3
**Contribution:** 2
**Rating:** 4
**Confidence:** 3

**Summary:**

Current MLLMs struggle with long event streams (from event cameras, which have microsecond precision but generate ~10⁶ timestamps/second) — either failing to interpret them or limiting to short sequences. To address this, the paper proposes LET-US, a framework that uses text-guided cross-modal queries to select salient event segments and hierarchical clustering to compress data (preserving key cues). Its two-stage training first boosts streaming data representation, then aligns event-text semantics. The authors also build EIQA-1M (1M+ event-text QA pairs for training) and EVQA-Bench (50k+ QA pairs for evaluation, timestamps up to 10⁹). Experiments show LET-US (Llama3.2-3B backbone) outperforms SOTA MLLMs on long event stream tasks.

**Strengths:**

1. Proposes LET-US, the first framework for long event stream textual alignment/understanding, handling up to 10⁹ timestamps.
2. Designs adaptive sequence reduction (cross-modal selection + clustering) to cut LLM inference cost while retaining key info.
3. Builds EIQA-1M (training dataset) and EVQA-Bench (evaluation benchmark for long event streams).
4. Validates LET-US outperforms SOTA (e.g., EventGPT, VideoLLaMA3) on classification, reasoning, etc., even with fewer parameters.

**Weaknesses:**

The reviewer’s primary concern centers on the experiment section, with specific feedback as follows:
1. Fairness of Baseline Comparisons: It appears that none of the baseline methods (e.g., EventGPT, VideoLLaMA3) are trained on the paper’s proposed datasets (EIQA-1M or EVQA-Bench), whereas the proposed LET-US framework is. This discrepancy creates an unfair comparison, as the baselines lack exposure to the same training data that enables LET-US to align event and text modalities—undermining the validity of performance comparisons.
2. Insufficiency of Ablation Studies: The paper only conducts ablation experiments on two components: Cross-Modal Guidance Compression (CMGC) and sampling strategies. However, additional critical ablation tests are needed. For instance:
Experiments to verify the effectiveness of the proposed datasets (e.g., comparing model performance when trained on EIQA-1M versus existing small-scale event datasets) are absent.
The dual-encoder design (SigLIP2 + DINOv2) is a key part of LET-US, but no experiments justify why this specific dual-encoder architecture is superior to single-encoder alternatives.
The necessity of the CMGC method is not sufficiently validated—there is no comparison with simpler compression approaches to demonstrate CMGC’s added value.
Beyond the experiment section, the datasets (EIQA-1M and EVQA-Bench) are framed as core contributions of the paper, yet their description lacks depth. More details (e.g., data collection pipelines, annotation quality control, statistical distribution across task categories) are needed to fully convey their novelty and utility for long event stream research.
Additionally, a typo was identified in the caption of Table 2: the term “Copression” should be corrected to “Compression”.

**Questions:**

See weakness.

---

### Official Review · Reviewer_VtGU · 2025-11-02

**Soundness:** 3
**Presentation:** 3
**Contribution:** 2
**Rating:** 4
**Confidence:** 3

**Summary:**

This paper introduces LET-US, a framework designed to help MLLMs understand long sequences of data from event cameras. Event cameras are special sensors that capture movement with very high speed and precision, but they generate a huge amount of data that is sparse and noisy, making it difficult for current AI models to process over long periods. The key idea of LET-US is to compress this long event stream by first using text queries to find important segments and then applying a clustering method to reduce redundancy. The authors also created two large-scale datasets (EIQA-1M and EVQA-Bench) that pair event data with text, which were previously lacking. Experiments show that LET-US performs better than existing state-of-the-art models across various tasks like classification, reasoning, and captioning on long event sequences.

**Strengths:**

- It tackles the challenging problem of enabling multimodal language models to understand very long and sparse event camera streams.

- The proposed compression method effectively reduces the long event sequences by using text queries and clustering to preserve key information.

- The creation of two large-scale event-text datasets is a significant resource that addresses a major data scarcity issue in the field.

- The model demonstrates strong performance, outperforming existing models across multiple tasks even with a smaller number of parameters.

**Weaknesses:**

- The process for generating the training dataset could be described with more transparency regarding quality control and potential biases.

- The comparison could be more direct with the few other event-based models, even on shorter sequences where they can run.

- The ablation study could be more comprehensive, for example by analyzing the impact of the two-stage training or the compression parameters (compression threshold τ and window size J)

- Some technical parts of the method could be explained with more intuitive descriptions to improve readability. For example, temporal compression formulas are quite technical. While not necessary to remove, adding a sentence or two in plain language explaining the intuition behind the clustering criteria (e.g., "bins that are very similar get merged aggressively") would improve readability.

**Questions:**

see weaknesses

---

### Note · Authors · 2025-11-14

**Comment:**

I sincerely thank the reviewers for their valuable feedback, which has provided significant guidance and greatly helped my work. I have become aware of several shortcomings in my current research. After careful consideration, I have decided to withdraw the submission in order to further refine my work.

**Withdrawal Confirmation:**

I have read and agree with the venue's withdrawal policy on behalf of myself and my co-authors.